# "Anybody can make kids; it takes a real man to look after your kids": Aboriginal men's discourse on parenting

Kootsy Canuto [1,2,3]*, Kurt Towers[4], Joshua Riessen[5], Jimmy Perry[6], Shane Bond[4], Dudley Ah Chee[4], Alex Brown[1,2,3]

1 Wardliparingga Aboriginal Research Unit, South Australian Health and Medical Research Institute, Adelaide, Australia, 2 Freemasons Foundation Centre for Men's Health, University of Adelaide, Adelaide, Australia, 3 Faculty of Health and Medical Sciences, School of Medicine, the University of Adelaide, Adelaide, Australia, 4 Watto Purrunna Aboriginal Primary Health Care Service, Adelaide, Australia, 5 Aboriginal Health Council of South Australia, Adelaide, Australia, 6 Aboriginal Drug & Alcohol Council (SA) Aboriginal Corporation, Underdale, Australia

* Kootsy.Canuto@sahmri.com

Data Availability Statement: The data that support the findings of this study are available for researchers who meet the criteria for access to confidential data upon request from the Aboriginal

## Abstract

### Background

The realms of parenting have long belonged to females. In many cultures it has been a female who has predominantly cared for and raised children. For many Aboriginal and Torres Strait Islander male parents this has resulted in them being largely overlooked from contributing to the parenting conversation. Predictably, such a dominant discourse has led to an inadequate distribution of opportunities available and a societal perception that Aboriginal and Torres Strait Islander male parents are disinterested in and/or disengaged from their parental roles and responsibilities, however, this is far from the truth.

### Methods

This study is entrenched in an Indigenist research approach which privileges Indigenous lives, Indigenous knowledges and Indigenous voices, and utilised the Research Topic Yarning method to capture participants stories.

### Results

Four yarning groups were conducted across South Australia in Coober Pedy, Yalata, Port Lincoln and metropolitan Adelaide. In total, 46 Aboriginal men contributed their experiences and stories of their roles and responsibilities as parents to this study.

Men described being a dad as a privilege, emotionally fulfilling and rewarding and although at times it can be challenging, neglecting their roles and responsibilities are not considered options. Lack of employment and therefore financial security were described as a challenge to fatherhood especially for fathers who live in remote communities. Aboriginal culture, connection to country and family were identified as critical elements and strengths for Aboriginal male parents. Furthermore, Aboriginal male parents are yearning for opportunities to participate in parenting programs including men's parenting groups.

Health Research Ethics Committee at Gokhan.
Ayturk@ahcsa.org.au. The data are not publicly
available due to ethical restrictions as they contain
information that could compromise the privacy of
Aboriginal or Torres Strait Islander research
participants. Imposed by the Aboriginal Health
Research Ethics Committee. For further details: Dr
Gokhan Ayturk (Senior Research and Ethics
Officer) Gokhan.Ayturk@ahcsa.org.au.

**Funding:** Financial support for this review was
received from the Lowitja Institute's Valuing
Aboriginal and Torres Strait Islander Young Men
Research Grant Funding, 2017 (research activity
017-YM-008), and the South Australian Health and
Medical Research Institute (SAHMRI) Women and
Kids Theme. AB is supported by a NHMRC Senior
Research Fellowship (APP1137563). Work
supported in part by NHMRC CRE (APP1061242).

**Competing interests:** The authors have declared
that no competing interest exist.

## Conclusion

Consideration of and concern for Aboriginal and Torres Strait Islander men's involvement and experiences prior to conception, prenatal and postpartum has slowly gained momentum in recent years, yet there has been little improvement in the overall provision of appropriate parenting support services and/or programs for these men.

## Introduction

Men in general have largely been overlooked from contributing to the parenting conversation [1] and 'until fairly recently, most research on family behaviour focused on women and children' exclusively [2]. With this in mind, it has been found that 'upholding or supporting rigid stereotypes can potentially hamper both girls' and boys' development' [3]. Predictably, such a discourse has led to an inadequate distribution of opportunities available for male parents to engage in, and a societal perception that these males are disinterested in and/or disengaged from their parental roles and responsibilities.

A systematic review by Baldwin & Bick (2018) of first time fathers' experiences and needs revealed that 'fathers wanted more guidance and support to prepare them for parenthood, specifically to better prepare them for subsequent relationship changes with their partner' [4]. Similarly, and in line with earlier research, Huusko and colleagues (2018) found that fathers in their study lacked the time during parental education classes to 'strengthen social support for first-time fathers' as they didn't have appropriate time to 'discuss with other expectant fathers since they had different needs and thoughts than mothers' [5].

In Australia, Western influence has dominated the maternal and child health care space with services giving 'centre stage to a women's passage through motherhood' [6]. Currently, there are a vast number of maternal programs whose focus is solely on mothers and their child whilst very few include fathers or are established exclusively for male parents [6, 7]. This theme also materializes within health research, where there is a plethora of studies concerning female parenting in contrast to that of male parenting [3, 8]. When the focus is on male parents, most 'studies to date have more often focused on the negative impacts of poor or absent fathering on children' [9]. Although there is considerable evidence regarding the positive influence fathers can have on their children's health, social success and academic achievements [10], the roles and responsibilities of male parents continue to be largely ignored by Australian maternal and early years services and research. The lack of emphasis placed on the important roles and responsibilities of Aboriginal and/or Torres Strait Islander male parents has led to regions of Australia where father-son and uncle-nephew relationships have been eroded [11]. Adams (2006) continues by suggesting 'the roles and responsibilities of Aboriginal elders, fathers, uncles, and grandfathers need to be emphasised and shared to strengthen family ties.' [11]. Disappointingly, the importance of family and family values within Aboriginal and Torres Strait Islander culture is rarely acknowledged, supported, and or understood by government authorities [12]. Similarly, Reilly and Rees (2018) suggest;

> Traditional Australian Aboriginal and Torres Strait Islander societies value men's role as parents; however, the importance of promoting fatherhood as a key social determinant of men's well-being has not been fully appreciated in Western medicine [7].

While it is noted that consideration of, and concern for, Aboriginal and Torres Strait Islander men's involvement and experiences prior to conception and throughout the prenatal

and postpartum period has slowly gained momentum in recent years, there has been little improvement in the overall provision of appropriate support services and/or programs for these men.

This study aimed to (i) explore the roles and responsibilities of Aboriginal and/or Torres Strait Islander male parents in South Australia as defined by men themselves and (ii) uncover strategies to assist how these men can be engaged appropriately to participate in the parenting realm to continue to be successful and supportive in their parenting roles.

Within this article, the term parent[ing] is interchangeably referred to as dad, guardian, carer and/or nurturer, as parenting is not just a biological role but one men assume as a social and cultural obligation [13]. The study took place in South Australia, which has a low proportion of the population who identify as Torres Strait Islander or Aboriginal and Torres Strait Islander, 4.5% of the total Aboriginal and Torres Strait Islander population, compared to 20% of the population in Queensland [14], although both Aboriginal and/or Torres Strait Islander men were the intended participants of this study, no Torres Strait Islander men participated in any of the four yarning groups, hence the title stating 'Aboriginal men' only.

## Methods

### Study design

A study reference group (SRG) made up of representatives from supporting organisations and services was established. The SRG members are Aboriginal and Torres Strait Islander people (men and women), including Aboriginal and Torres Strait Islander male community members and one non-Indigenous male. The Aboriginal and Torres Strait Islander men of the SRG guided the research team throughout the research process; their guidance ensured the research was conducted appropriately.

To better understand Aboriginal and Torres Strait Islander men's perceptions, expectations, and aspirations of successful parenting it was important to document their narratives. Through these narratives, we would be able to comprehend what these parents needed to strengthen their social and emotional wellbeing thus enabling them to perform their roles and responsibilities as parents. The study utilised the framework and domains of Social and Emotional Wellbeing put forth by Gee et al which depicted the domains of wellbeing that typically characterise social and emotional wellbeing from an Aboriginal and Torres Strait Islanders' Perspective [15]. This study is entrenched in an Indigenist research approach which privileges Indigenous lives, Indigenous knowledges and Indigenous voices [16].

This study uses a yarning research process called Research Topic Yarning, of which the sole purpose is to 'gather information through participant's stories that are related to the research topic. While the yarn is relaxed and interactive it is also purposeful with a defined beginning and end' [17]. Geia, Hayes & Usher (2013) support such a process and contend that 'insights from Indigenous persons/cultures can contribute to our understanding of storytelling or yarning as a methodological research practice in relation to Indigenous peoples and their communities' [18]. Furthermore, Dean (2010) explains how 'Indigenous research should reflect the authority and foundations of Indigenous knowledge systems and yarning as a methodology can permit this' [19].

The research was led by a Torres Strait Islander man (KC), who is an experienced qualitative researcher. The yarning groups were co-facilitated by KC and an Aboriginal male health worker and health service team leader (KT). KT led the yarning group discussions using a semi-structured yarning guide. At the beginning of each yarning group, the co-facilitators spoke about confidentiality, privacy and the proposed nature of the discussion, explaining that the group is there to discuss parenting from an Aboriginal and/or Torres Strait Islander male perspective. At each yarning group at least one Aboriginal male social and emotional wellbeing

health worker was present and available for participants if needed before, during or after the yarning groups. Aboriginal male health workers were also invited to participate in and observe the yarning groups sessions as a capacity building opportunity for them and to assist in community engagement as they were well known and respected Aboriginal men in South Australia. This study provided the research team with opportunities to learn from each other and all Aboriginal male health workers and social and emotional wellbeing workers engaged in the project were active research team members, with each attending at least one yarning group and as such are authors on this manuscript.

As explained by Tong, Sainsbury and Craig (2007) within the comprehensive Consolidated Criteria for Reporting Qualitative Research (COREQ) checklist, it is important to let the reader understand the personal characteristics of the researcher and/or research team and the relationship the researcher and/or team have with the participants [20]. As mentioned previously, lead author (also lead researcher of this study) KC is a Torres Strait Islander man with his own children and several nephews and nieces for who he provides parenting to. The remaining members of the research team are all Aboriginal men who work within Aboriginal and Torres Strait Islander health and have children of their own. The fact that the researchers were all Aboriginal or Torres Strait Islander fathers allowed for common ground with the participants and an ability to have far deeper and more frank conversations with the participants before, during and after the recorded yarning group sessions. At each of the yarning group sites, at least one member of the research team either had an established relationship with some or all of the participants which created favourable and safe interview conditions. Tong et al. (2007) explains how;

> The relationship and 'extent of interaction between the researcher and their participants should be described as it can have an effect on participants' responses and also on the researchers' understanding of the phenomena [20].

## Data collection

The study was approved by the Aboriginal Health Council of South Australia's Aboriginal Health Research Ethics Committee (04-18-785).

## Participant recruitment

All participants were invited to participate opportunistically at locations deemed suitable by the participants and at suitable times according to community protocols; for example, avoiding cultural ceremony or sorry business (death in the community). To be included the participants had to be male, identify as Aboriginal and/or a Torres Strait Islander, aged 16 years and over, were parents or provided nurturing and/or caring roles and responsibilities, resided in one of the four catchment areas of Yalata, Coober Pedy, Port Lincoln and metropolitan Adelaide, and were able to provide informed written consent. It should be noted that no minors (under 18 years) were involved in this study. As a token of appreciation, a $50 gift voucher was presented to the men for their input, time and knowledge. Protocols relating to the collection, use, management and storage of data were developed by the SRG, in accordance with the South Australian Aboriginal Health Research Accord [21] and adhered to by the research team. This was a small pilot study. Available resources limited the study to only four yarning groups, one per community.

## Yarning group locations

The Port Lincoln yarning group was conducted at the Port Lincoln Aboriginal Health Service (PLAHS) where many of us from the research team had a rapport with staff and some of the

local men. We knew the men of the community were comfortable coming to PLAHS for a yarning group.

The Coober Pedy yarning group was conducted at the Umoona Tjutagku Health Service Aboriginal Corporation where they serve a daily breakfast for many members of the community. The men were more than willing to participate at this setting.

Due to sorry business, the yarning group with Yalata participants was conducted at the Wahgunyah Conservation Reserve, 90-minutes' drive from Yalata, as requested by the men, creating a safe space for deep conversations.

The Adelaide metro yarning group was conducted at the Aboriginal Health Council of South Australia (AHCSA) where the research team and participants all had a rapport with the service and some of the staff.

### Informed consent

Prior to each yarning group, the study was explained to the participants with the aid of a participant information sheet and verbally by the co-facilitators. It was estimated that each yarning group would take approximately one hour to complete. Informed consent to audio record, transcribe and analyse the data from each of the yarning groups was obtained from every participant. Although English is not the first language of the Aboriginal male participants in Yalata, members of the research team (KT and JP) have strong cultural connections with the community and it was deemed not necessary to have an interpreter as their conversational English is sufficient to have meaningful conversations. Yarning group recordings were transcribed for analysis by KC.

### Yarning group prompt questions

All four groups were asked the same semi-structured broad, open-ended questions which focused on their roles and responsibilities as parents, nurturers and/or carers whether they were fathers, uncles, grandfathers or within their role within the kinship system. Some of the questions asked were; what keeps you strong as parents?; when it comes to caring for your child how do you lead and overcome challenges? and what do you think you need to be successful or to continue to be successful in your role? Participants were also asked how services can make being a dad, uncle, grandparent or carer easier for them.

### Data analysis

Data collected from the yarning groups was analysed using thematic network analysis which extracts themes into three distinct categories; basic themes, organising themes and global themes [22]. Thematic network analysis was chosen because the data collected can be represented as a web-like map which illustrates the relationship between the three categories [22]. This process was completed by KC without the use of computer software. KC, SB, DA, JR all listened to the recordings and discussed the coding of the data until consensus was reached. KC, KT & JP, who were all present at each yarning session, clarified all meanings and quotes from parts of the audio recording that were difficult to hear.

### Results

Participants were recruited from Coober Pedy (n = 12), Yalata (n = 12), Port Lincoln (n = 17) and Adelaide metro (n = 5). The four yarning groups ran for an average of approximately 49 minutes, with the longest being 1 hour and 14 minutes.

From our experience in this study it was evident how the teams identity, credentials, occupations, gender, experience and training influenced how the yarning groups were performed and how the participants from each of the community supported and interacted with the research team [20].

From the thematic analysis four global themes were identified; being a dad, employment, cultural and community connections and opportunities to participate. The analysis including the organisational and local themes are outlined in Table 1. Within and across these themes

**Table 1.**

| Global Themes | Organisational Themes | Basic Themes |
|---|---|---|
| Being a Dad | Support | • Be there for your child(ren)<br>• Be involved in your child(ren)'s life<br>• Give your child(ren) direction<br>• Being a protector and provider<br>• Give your child(ren) what you never had–a better life<br>• Being a guide, a teacher, and sharing your wisdom you've learned in life |
| | Challenges | • Increased responsibilities as a parent<br>• Raising daughters can be difficult for a dad especially throughout their pubescent years<br>• Put food on table/pay the bills<br>• Put oneself into a position of financial security<br>• Keeping up with technology is a challenge<br>• We need something for our kids to do–especially after the footy season ends |
| | Knowledge | • When and where male parents can access the necessary tools to support them in their roles<br>• Finding alternate solutions to engage and support children in the ever-advancing technological age |
| Employment | Lack of jobs | • Need to have local, appropriate employment opportunities |
| | Unsuitable work | • Fly in, fly out work (away from home for long periods)<br>• A lot of employment opportunities require numeracy and literacy skills |
| Cultural and Community Connections | Family and Kinship | • Although I have no kids, I still have a fatherly role<br>• Does not matter what tribe you are from, we are all one<br>• Parenting is not an individual's job–the whole community is required<br>• Family provide support and give you strength |
| | Connections | • Being on country is important<br>• Knowing who you are and where you are connected is very important |
| | Respect | • We need to teach our children respect and we need to teach them culture<br>• Elder's stories are not being told. We are trying to show our kids the traditional guidelines on how to do things properly |
| | Role Modelling | • How my father (and uncles) treated me moulded me into the person I have become<br>• Move away from the stereotypical negative narrative. Set a positive example<br>• Normalise the role of a father and uncle in a child's life<br>• If you treat your child good your child treats their kids equally good<br>• Kids do what their father's do<br>• Fathers make good role models for their children |
| Opportunities to participate | Dedicated Men's Space | • Providing a men's parent group so we can communicate and share concerns<br>• Having somewhere for men to come together and talk more as fathers<br>• Being able to have a yarn in support sessions about being a dad and what to expect; learning from other dads<br>• Having something (program/space) permanent. Not something that is run a couple hours a week and you go away and that's it; something every day |
| | Services/ Organisations | • Important for services and organisations to recognise that dads play an integral role in a child's life–not only mums<br>• Hospital staff to be more respectful and acknowledge the significance of your child being born<br>• There's a lot of programs for mums and bubs and families–but there is none dedicated to dads<br>• Create culturally appropriate opportunities for men to participate throughout the birthing and parenting process<br>• Understand Aboriginal males deal with a lot of added pressure and expectations (positive and negative) in society<br>• Be more conscious of the way you, as individuals and organisations, interact with Aboriginal men |

are identified roles and responsibilities of being a dad/male parent; some of the barriers and enablers influencing men's ability to fulfil these roles and responsibilities; and potential strategies to improve engagement of men through appropriate services to support their parenting roles.

## Being a dad

All four yarning groups were asked what it means to be a father and although all men spoke of the positive elements of fatherhood, they also acknowledged a number of challenges having a child to care for involved, "I love my kids and I will continue to love my kids for as long as I am alive". (Adelaide metro participant). "You still worry for your kids regardless of what age they are, it's just a natural thing". (Yalata participant). "The principle of being a father is to protect your family and provide the things that they need to have a happy and healthy life". (Adelaide metro participant). "It's harder for us as a male especially if we have daughters, especially when they go through their periods of life and their development, how are we as males supposed to give them [our daughters] advice"? (Port Lincoln participant). For some of the male parents it was just being present in their child's lives that brought them the most satisfaction in their roles;

> Just watching them succeed in life is the best, to be there to watch them, to grow as individuals and succeed, it doesn't have to be huge things, watching my daughter at her year-12 formal; beautiful woman, she's growing into a beautiful woman . . .as a father there are so many things, but just watching them grow and being successful in their own little ways, it doesn't have to be winning the Nobel Peace Prize, but those little things have been the best for me. (Port Lincoln participant).

Moreover, the role of male parents to support their children was clear throughout the four focus-groups, "Supporting kids, especially the young to go to school". (Yalata participant). One parent believed that staying healthy was important to be a dad, "fundamentally it's about being alive, fit, strong and healthy so you're there for your children, nieces and nephews" (Adelaide metro participant). Also, "I just want to be there with my kids". (Yalata participant). An Adelaide metro participant summed it up;

> Being there for your children, especially through their milestones, sharing in their experiences as they grow up, being there when they hopefully graduate, guiding them through school, being there for their birthdays, under any circumstances try to be there in their life, have a presence in their life. (Adelaide metro participant).

Being present, involved and there for their children under any and all circumstances were echoed by most of the men throughout the four focus-groups across all four regions. For example, a Yalata participant stated that "it's quite easy to be a father, but you've got to be a dad as well". (Yalata participant). This reaffirmed that these parents understood the importance and significance of their role in the family unit and as parents. As one Port Lincoln participant describes;

> We all know what we want for our kids and we all struggle to get that for our kids and it's all positive. From my perspective, it's our creations, we bought those lives into this world and it's our responsibility to make sure we give them a start to life. Developing them and watching them grow from this age [young] and giving them the tools, like we spoke about, then we wrapped around that nurturing, love and caring that we do actually care about them and try to point them in the right direction . . .it's just a matter of guidance, love and

care, and knowing that they know you are there for them through thick and thin. (Port Lincoln participant).

However, it was acknowledged that being a dad was not always the easiest role to perform. As one participant explained;

I remember having times when I thought I can't do this [being a dad] . . .but you get through it, everybody does. It [parenting] is an emotional rollercoaster of highs and lows and worries that you've never had. You worry sometimes about everything and stupid stuff, but that's just probably all part of it. I think one of the challenges is trying not to let that get in the way of your parenting because when you're stressed or you're angry about stuff happening with your relationship, or anything in your life, you don't want that to come in to your interactions with your children. However, it's really hard to separate sometimes stuff happening outside of your family and stuff happening inside of your family. (Adelaide metro participant).

For another Adelaide metro participant, being a dad was like being a spectator at times;

I feel like I did a lot of things around the care of the babies, but it wasn't actual providing direct care. I still changed nappies and stuff like that, but I was more sort of doing stuff around being supportive, whereas the mum was doing the more of the direct stuff. I did more of the stuff around the edges . . . Both my kids breast fed as well, and it was almost like they were attached to their mum all the time. I would go to work, and it wasn't until they were one or so when they come to the next sort of phase of development that I could even get a closer bond. Not that you don't love them or anything, but it's something about when they were babies they were these little people that are so dependent on their mums I feel for everything, and when they get a bit older the dads can have a bit more involvement in doing the care and feeding. (Adelaide metro participant).

What it meant to be a dad varied amongst the men. For some, it was a privilege, emotionally fulfilling and rewarding, especially as the father's role changed over time;

Being a guide and a teacher, sharing your wisdom you've learned in life and try to guide your kids in the right way . . .at times I was a mentor, I was a coach and I was a teacher [to my kids]. Being involved in their lives and taking an interest and showing that they matter to you. I've found that it is paying off big time for me now that they are adults. It's really great having conversations with them now. It's kind of like having your own best friends. (Adelaide metro participant).

For some, it was a challenge. However, for all the men, neglecting their roles and responsibilities as parents were never considered options;

I want to change the [negative Aboriginal man's] narrative in the child's life. I want others to see a man who has a job and he is behaving decently . . .setting the example. (Adelaide metro participant).

## Employment

Without employment it is difficult to afford the basic necessities to raise a child and to provide for the whole family unit. In Coober Pedy, employment was identified as a major challenge to performing the roles and responsibilities of these male parents;

My job is to go to work, to be there for them to put food on the table to pay bills. If you take that away from a bloke then what have you got, unemployment. They [employment services] are not going to fix it, they not fixing it for the last twenty years . . .I nearly worked all my life, but without work how you meant to deal with, you gonna hit rock bottom and you gonna feel it . . .we work, we wanna go to work, give us something [employment] to be happy to go to work. (Coober Pedy participant).

When these males are afforded employment opportunities they are then confronted with complex challenges. As one participant explained;

There are jobs out of town . . .but do you want to work away from family for five years, five years in the mines doing twelve-hour shifts? . . .it took me a while to get a job back in this town . . . there's no jobs here [Coober Pedy] . . .when you work away from family that's when your mind starts working overtime, you start stressing out, money is coming in but I'm working away from my family. (Coober Pedy participant).

Employment was noted throughout all four yarning groups as essential to contributing to the health and wellbeing of families. For example, "putting oneself into a position of financial security" (Port Lincoln participant) was important to some of the male parents. For one Yalata male parent, employment, along with connection to country, was a catalyst for positive change;

Participant: A father is everything, I lost my father at a young age [10 years of age] and it hurt me . . .when my father passed, I had nothing left for me and I moved to sniffing petrol for a while then I moved to the city and starting getting into crime in Adelaide and could of ended up in jail, but charges got dropped then I ended up having a son so I moved back home for work and now I got two boys and I am still working and I haven't stopped working [since 2009].

Facilitator: What made you be a better person/dad?

Participant: Being back on country. (Yalata participant).

## Cultural and community connections

The participants all commented on how culture was a critical element to successfully performing the roles and responsibilities as parents. Although cultures across Australia's Aboriginal and Torres Strait Islander people differ greatly for example in language, customs, beliefs and traditions there are many commonalities shared, such as the importance and connection to place. Throughout the four regions, connection to culture, country and family was essential. In Coober Pedy, teaching the children respect and culture was important and so was challenging how technology was encroaching on culture. This challenge meant alternate strategies were being developed to continue to embrace culture;

We grew up and there was no technology that is around now . . .we grew up out bush hunting . . .we trying to find other ways and strategies for our kids to grow up proper way, out of town teaching them the ropes to survival, getting them away from town but also teaching. We as leaders [are] trying to show them how, to encourage their friends and their cousins, and they go back to talk to their parents to give them a little nudge to help them to take them out bush to find plants, to keep them motivated, take them out fishing. (Coober Pedy participant).

Although technology was challenging for some of the male parents with regards to infringing on culture, it was also seen as beneficial, "technology is good for our kids because they learn to read and they keep up with their technology, we [parents] not keeping up with them, we are behind . . .I find it [technology] good in a way but bad too". (Coober Pedy participant).

For an Adelaide metro parent family was important;

> family do support you and they do give you strength in times of at your lows and stuff. I think knowing I'm going to see my kids again and everything I do is for my boys, absolutely everything, so that gives me strength and I give that back to them. It comes back to being a good role model. I draw a lot of strength from just thinking about them every day and everything I do". (Adelaide metro participant).

In Port Lincoln one of the participants highlighted how "knowing who you are, and where you are connected, is very important" (Port Lincoln participant). For one of the participants living away from his country in metropolitan Adelaide he longed for connection to culture for his children;

> I want my kids to be linked in with all my family and then that culture as well. I want my kids to have opportunities to learn more culture and ways that I can support that and do that with my family more as well . . .Opportunities to do that when you are not there with your kids can be a bit harder as well. (Adelaide metro participant).

As demonstrated, culture is viewed as critical to the development of the child and strengthening of the family unit. Culture is also important when it comes to how other males participate in the role of parenting. In Port Lincoln participants spoke about the roles of non-biological fathers, "I don't have any kids, but I still have a fatherly role". (Port Lincoln participant). Furthermore, Coober Pedy male parents agreed that, "it does not matter what tribe you are from; we are all one". (Coober Pedy participant). One participant stated that, "our brothers and nephews play the roles as fathers too . . .even if they uncles they play a part, they have a responsibility, they're dads too, even the grandfathers". (Coober Pedy participant). This sentiment was echoed by a Yalata participant who explained that, "uncles have a very important role, as in respect, for their nieces and nephews to bring them up with respect". (Yalata participant).

Being present at the birth of the child formed a stronger family bond and appreciation from the father's perspective, "it changes you seeing the pain the mother goes through [child birth] . . .I was so drained afterwards but not as much as she was [mother] . . .made me appreciate her [mother] more". (Adelaide metro participant). Also, transgenerational behaviour effects parenting and the child's ultimate outcomes as a Yalata participant explained, "becoming a grandfather is a whole new ballgame, everything changes, do the right thing for your kids and they will do the right thing for their kids". (Yalata participant). This was echoed by a Port Lincoln participant, "give them [children] the tools to move forward . . .to live their own life and when they're old enough, hopefully, what you taught them now, they keep in their heads". (Port Lincoln participant). Notably, challenging the stereotypical culture of parenting was seen as necessary, and could be achieved by recognising;

> The importance that dads play [in parenting] and [understanding] it's not just the mums [who] look after kids. We have an important role to play and having spaces and a chance to come together and talk [is critical]". (Adelaide metro participant).

## Having an opportunity to participate in the parenting process

The Aboriginal male parents who participated in this research all appreciated the invitation and opportunity to participate in this research. They were all happy to speak about the roles and responsibilities they have as parents. Without doubt, being included in the parenting process was important to these males;

> Create opportunities for men . . .it would be good for men to have [the opportunity to learn more], you're about to be a dad check, let's have a yarn about what kind of dad you are going to be and what's involved in being a dad, because I never had anything like that, and you don't know anything I guess when you are a new dad for the first time. You are really just learning as you go. (Adelaide metro participant).

In the same yarning group, another Adelaide metro participant explained that;

> . . .men [should] come together and talk more. As the father aspect, we need to normalise that a bit more . . .and we would benefit from that I reckon as well. It is bloody good talking about things, and you do feel better when you talk about things and heard stories from other people. (Adelaide metro participant).

Having a space for Aboriginal and Torres Strait Islander male parents to come together was also echoed by a Port Lincoln participant who stated how;

> We need something that needs to be permanent, not something that is run just a couple hours a week and goes away and that's it. Somewhere everyday . . .a hub for the males to come to that is always available. (Port Lincoln participant).

Again, as one participant explained, the importance of a safe place to talk and connect was important;

> Providing a space like this [men's parent group] to be able to communicate . . .it's so refreshing, and it helps you to understand what you are going through is shared with many others and yeah, it takes a bit of weight of your shoulders. (Adelaide metro participant).

Similarly, in Port Lincoln, one male parent was concerned for other male parents "more support like this session should be more regular for young fellas around here, there are more like me, men who should be here to know what's going on". (Port Lincoln participant).

Many men from all four yarning groups acknowledged the plethora of maternal programs and support offered and the lack of programs for fathers, "there's a lot of programs for mums and bubs and families . . .and they're good things, I'm not dragging them down, but where's the dads voice in there?" (Adelaide metro participant). Further, although the male parents from Adelaide metro and Port Lincoln had plenty to say about what they needed in way of access to and opportunities to participate and come together, for some of the Yalata participants, they were more focussed on "needing something for the kids to do". (Yalata participant). Whilst, for some of the Coober Pedy participants, they highlighted support from their community as being important, "we need the support from the whole community to all work as one".(Coober Pedy participant).

Finally, when the men spoke about the services they did receive, one participant explained his disappointment at the lack of empathy and support he received during the birth of his child;

Acknowledging that this is not your 30,000<sup>th</sup> baby for you. This is actually something so special and that it is hard, and it is full on and you don't know what you're doing. Just to be really respectful, friendly and supportive and have some empathy and all that kind of stuff and not just be like get it out of the way and the next one comes in. (Adelaide metro participant).

Despite gender, someone having feelings of exclusion from any emotional and practical support during this critical time is cause for concern.

## Discussion

This study summarises the roles and responsibilities of Aboriginal male parents as defined by the men themselves. This study also outlines the many challenges these men face as Aboriginal men entering parenting for the first time or of those currently involved in the parenting realm. The results of our study correlate with the limited literature surrounding Aboriginal and Torres Strait Islander male parenting; highlighting the need for maternal and early years services to improve their understanding of the needs and desires of Aboriginal male parents and provide culturally appropriate services to these men.

The majority of parenting services are designed for and focused on female parents; and as previously mentioned, for good reason. However, the roles and responsibilities traditionally bestowed upon female parents from all sectors of Western society has led to male parents being largely excluded from the parenting conversation. In their report, *Engaging fathers*: *Evidence review*, Fletcher et al. (2014) highlights the importance of inclusivity within the parenting domain;

Research clearly demonstrates that without deliberate measures to develop and incorporate father-inclusive practice into all aspects of services delivery, service providers will overwhelmingly concentrate their efforts on maternal/child dyads. Research also shows this may result in both lower levels of satisfaction for families using the service and poorer outcomes for children, mothers and fathers [10].

Although the journey to full inclusivity within maternal and early years services will take time, it should be noted in recent years the parenting domain is beginning to consider the roles, responsibilities and value male parents can add prior to conception and through the birthing and parenting journey. This change in approach is due to the 'mounting evidence supporting the critical role of fathering' [9] and community demand.

Whilst there has been a rise in male parents wanting to be included in the parenting realm, the voices of Aboriginal and Torres Strait Islander men still remain on the margins and largely unheard. As positive as this is for male parents in general, the roles and responsibilities of Aboriginal and Torres Strait Islander male parents remain complex and underexplored. What is clear, is the 'importance of creating a father-friendly environment' [23] for Aboriginal and Torres Strait Islander men.

This study has revealed, given the opportunity, Aboriginal and/or Torres Strait Islander male parents are interested and committed to their roles and responsibilities as parents. Unfortunately, appropriate programs and services remain out of reach for some Aboriginal and Torres Strait Islander male parents as the *Aboriginal Dads Program* in Port Augusta, South Australia, noted, 'there are no services offering support and education to new Aboriginal fathers' [24]. The importance of having appropriate resources for Aboriginal and Torres Strait Islander male parents has been well documented [11, 23, 25–27], however, the motivation and ability to adhere to these wants and needs remains lacking.

Furthermore, this study highlighted the want and need of Aboriginal men to be involved in the parenting process from pre-natal, during pregnancy, post-partum and throughout the child's life. As previously mentioned, the changing role of fathers across the different stages of their child's growth is important to recognise and support. Subsequently, providing appropriate education and support services that can allow Aboriginal and Torres Strait Islander men to perform their roles and responsibilities to the best of their ability is crucial for the whole family.

The strengths and limitations of this small qualitative research study need to be considered. One of the key strengths of this study is the research team are all Aboriginal and/or Torres Strait Islander male parents, which assisted with the team's engagement with the participants in deeper, more meaningful conversations and their contextual knowledge for the interpretation of the discussions.

Although yarning groups were conducted across urban, remote and very remote locations, due to the small sample size it may not be representative of all Aboriginal male parents across Australia. In addition, demographic information was not collected from participants limiting our ability to determine the contextual factors that may influence their responses, such as their relationship status or their current employment status.

The study participants may be a bias sample as the yarning group topic was advertised in advance. In addition, the group setting may have deterred participants from expressing opinions that might have been deemed negative or have gone 'against the grain', which is a common phenomenon in group interviews. In addition, as mentioned previously, no Torres Strait Islander male parents participated in the study.

It should be noted that opportunities to translate the study's findings into sustainable changes in policy and/or practice continue to be explored by members of the SRG and the research team.

## Conclusion

The lack of parental education, support programs and services that exists for Aboriginal and Torres Strait Islander male parents along with a societal perception that undervalues the importance of these parents clearly demonstrates a damaged system. In saying this, any shift of the social norms and stigmas regarding Aboriginal and Torres Strait Islander males parenting will take significant time, in the meantime maternal and early years services 'should acknowledge the local stigmas that exist and work with local Indigenous men to find ways to make accessing service more appropriate' [28].

As this study revealed, maternal and early years services should do more to better engage Aboriginal and Torres Strait Islander male parents. Although there is limited opportunity for these parents to fully participate in and contribute to such services, when given the opportunity, Aboriginal and Torres Strait Islander male parents are willing to engage with and contribute to appropriate opportunities of inclusivity.

The third report of the *State of the World's Fathers*: *Unlocking the power of men's care*:*2019*, focused on equality in unpaid care work around the world and suggests;

> . . .by mobilizing new laws and policies, shifting social and gender norms, promoting family and relationship equality, and empowering individual men and fathers to become equitable caregivers, the world will see radical transformations [29].

This report highlights the importance of including male parents into the realm of parenting and argues for equality between male and female parents in all aspects of child rearing and

parental responsibilities for the benefit of the whole family and the community. However, for Aboriginal and Torres Strait Islander people, being a marginalised minority of a colonised country, achieving equity and equality continues to be a challenge. Therefore, the best Aboriginal and Torres Strait Islander male parents can hope for, right now, is to be included into the conversation regarding parenting.

It is evident that more should be done to support Aboriginal and Torres Strait Islander male parents, and this should start with the acknowledgment of our vital roles in the healthy development of children and our families. This study uncovered the roles, responsibilities, wants and needs of Aboriginal male parents and unfortunately, the exclusion of Aboriginal and Torres Strait Islander men from the parenting domain highlights a birthing and early years system that is broken. Without urgent correction, this will be to the detriment of these male parents, their children, families and communities.

# Acknowledgments

Thank you to the Aboriginal and/or Torres Strait Islander men who participated in this study. We would like to acknowledge the custodians of the land on which the interviews took place for this study; the Kaurna people, the custodians of Adelaide and the Adelaide Plains, the Wirangu people, the custodians of the Far Western coast of South Australia, the Banggarla people, the custodians from the Northern region of Spencer Gulf up to the south of Port Lincoln, and the Antakarinja people, the custodians of the Coober Pedy district.

Thank you to Amanda Mitchell and Karen Glover for all the work you did behind the scenes, including the effort you both put into getting the Strong Dads Strong Futures project off the ground.

KC would like to acknowledge the significant administrative support provided from the Lowitja Institute, the Wardliparingga Aboriginal Health Equity Unit within the SAHMRI, the Aboriginal Health Council of South Australia (AHCSA), SAHMRI Women and Kids Theme, and the Freemasons Foundation Centre for Men's Health, University of Adelaide.

# Author Contributions

**Conceptualization:** Kurt Towers, Joshua Riessen, Jimmy Perry, Shane Bond, Dudley Ah Chee, Alex Brown.

**Data curation:** Kootsy Canuto, Kurt Towers, Joshua Riessen, Jimmy Perry, Shane Bond, Dudley Ah Chee, Alex Brown.

**Formal analysis:** Kootsy Canuto, Kurt Towers, Joshua Riessen, Jimmy Perry, Shane Bond, Dudley Ah Chee, Alex Brown.

**Funding acquisition:** Kootsy Canuto, Alex Brown.

**Investigation:** Kootsy Canuto, Kurt Towers, Joshua Riessen, Jimmy Perry, Shane Bond, Dudley Ah Chee, Alex Brown.

**Methodology:** Kootsy Canuto, Alex Brown.

**Project administration:** Kootsy Canuto, Kurt Towers, Joshua Riessen, Jimmy Perry, Shane Bond, Dudley Ah Chee.

**Resources:** Kurt Towers, Joshua Riessen, Jimmy Perry, Alex Brown.

**Supervision:** Kootsy Canuto, Alex Brown.

**Writing – original draft:** Kootsy Canuto.

**Writing – review & editing:** Kurt Towers, Joshua Riessen, Jimmy Perry, Shane Bond, Dudley Ah Chee, Alex Brown.

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
