## [Decision Letter · Decision Letter 0]

9 Sep 2019

PONE-D-19-21777

“Anybody can make kids; it takes a real man to look after your kids”: Aboriginal men’s discourse on parenting.

PLOS ONE

Dear Dr Canuto,

Thank you for submitting your manuscript to PLOS ONE. After careful consideration, we feel that it has merit but does not fully meet PLOS ONE’s publication criteria as it currently stands. Therefore, we invite you to submit a revised version of the manuscript that addresses the points raised during the review process.

In particular, attention is needed to providing and clarifying context and population particularly for a global readership who may be less familiar with this community. In addition, it will be helpful to streamline and clarify methods and sources of evidence from the data collection. The peer reviewers have provided specific comments to guide these revisions. 

We would appreciate receiving your revised manuscript by Oct 24 2019 11:59PM. To enhance the reproducibility of your results, we recommend that if applicable you deposit your laboratory protocols in protocols.io, where a protocol can be assigned its own identifier (DOI) such that it can be cited independently in the future. For instructions see: http://journals.plos.org/plosone/s/submission-guidelines#loc-laboratory-protocols

We look forward to receiving your revised manuscript.

Kind regards,

Aisha K Yousafzai

Academic Editor

PLOS ONE

Journal Requirements:

2.  Please include a copy of the interview guide used in the study, in both the original language and English, as Supporting Information, or include a citation if it has been published previously.

3. You indicated that you had ethical approval for your study. In your Methods section, please ensure you have also stated whether you obtained consent from parents or guardians of the minors included in the study or whether the research ethics committee or IRB specifically waived the need for their consent.

Additional Editor Comments (if provided):

Reviewers' comments:

Reviewer's Responses to Questions

**Comments to the Author**

1. Is the manuscript technically sound, and do the data support the conclusions?

Reviewer #1: Partly

Reviewer #2: No

2. Has the statistical analysis been performed appropriately and rigorously? 

Reviewer #1: N/A

Reviewer #2: N/A

3. Have the authors made all data underlying the findings in their manuscript fully available?

Reviewer #1: No

Reviewer #2: No

4. Is the manuscript presented in an intelligible fashion and written in standard English?

Reviewer #1: Yes

Reviewer #2: Yes

5. Review Comments to the Author

Reviewer #1: This manuscript reports findings from a study with Aboriginal men residing in Australia. Specifically, the authors utilized an Indigenous cultural form of conversation (i.e., yarning) as a data-gathering tool and found several themes centering around the perceived role as a male parent as well as the motivation to be engaged in parenting programs. There are certain things to like about this study. For instance, it reports findings from a population that is not well-represented in the literature and for which our knowledge of paternal engagement in child-rearing is limited. However, there are several concerns that the authors may need to address as listed in the following paragraphs.

1. There are various direct quotes for which the page number should be added for complete and adequate reference.

2. Could the authors comment a bit more on the reasons that led to the lack of Torre Strait Islander males participating in any of the yarning groups? The SRG comprised both Aboriginal and Torres Strait Islander people, which should have facilitated recruitment of Torres Strait Islander.

3. One goal (stated in the Study Design section) was to comprehend what parents needed to strengthen their social and emotional wellbeing. Please provide an (operational) definition of social and emotional wellbeing and related questions to elicit such narratives.

4. How is “remote father” defined (Abstract)? Does “remote” refer to the study location? If so, the findings reported in the Results do not clearly describe the similarities and differences between the narratives of remote and non-remote fathers.

5. Some information should be moved to the Data Collection section. For instance, move the information about the language in which the yarning groups were conducted (p. 9, lines 210-213) from the Results to the Data Collection section. The information about the study locations (p. 9, lines 215-227) would also be better placed in the Data Collection section.

6. Elaborate on the factors that determined data saturation. The interview guide/questions are rather limited with only two questions (“What keeps you strong as a parent?” and “What do you think you need to be successful or to continue to be successful in your role?”). Perhaps the researchers could have elicited additional information by using relevant follow-up questions. Also, how was “success” (or, being successful) defined for the second question?

7. Which considerations guided the determination of the group size? While the groups were fairly balanced in the first three locations, there were only 5 men in the last group (Adelaide metro).

8. It is important to present the key demographics of the participants. For instance, how old were the participants? How many were the biological fathers of the children they provided care for? How many children did the participants have and how old were they? Were all participating fathers married to the mother of their children? How many fathers were employed and other mothers employed as well? What were the household compositions and living arrangements of the participating fathers? Such information would greatly help in contextualizing the findings and themes (e.g., the finding that lack of employment and financial insecurity were described as challenges to fatherhood, or the importance of the family unit and the role of non-biological fathers). There currently are several instances in which the information would benefit from additional nuances and a more in-depth discussion of the sample characteristics. For instance, the second sentence of the Discussion states that “this study also outlines the many challenges these men face as Aboriginal men entering parenting for the first time or of those currently involved in the parenting realm.” Again, it is unclear how many men are first-time fathers and how many participants already provide care to children (and in which caregiving arrangement).

9. Related to the comment above, there is inconsistency concerning the sample description. For instance, on page 5, lines 108-110, the authors clarify that no Torres Strait Islander males participated in the yearning groups. Later, on page 17, line 402, it is stated that “the Aboriginal and Torres Strait Islander male parents who participated in this research all appreciated the opportunity to participate in this research […]”. Clarify and be consistent in whether Torres Strait Islander males participated in this study. Again, a detailed description of key demographics would help in this regard.

10. There is a reference to thematic network analyses to illustrate the relationship between the three categories. Why did the authors decide not to present these findings (“web-like map”) on basic themes, organizing themes and global themes? Doing so would help outline how the four categories (i.e., “Being a dad”, “Employment”, “Culture, country and family connectedness” and “Having an opportunity to participate in the parenting process”) were derived from the interview material.

11. Culture is a dominant term used throughout the Results section. To facilitate the understanding of the key findings, briefly define “culture” either based on the literature or as defined by participants who referred to culture in their narratives.

12. The Introduction and the Discussion sections would benefit from placing the study aims and objectives into the broader research context. For instance, Michael Lamb, Joseph Pleck, and others have developed and continuously revised conceptualizations of paternal involvement. How do the findings of this study contribute to these (and related) conceptualizations?

13. Some statements and conclusions in the discussion overstretch the study findings. For instance, how can the authors be sure that the system was designed “without them in mind” (page 21, line 501)? I encourage the researchers to revise such statements according to the explanatory power of their data as well as the limitations of their study.

14. Minor proof-reading required (e.g., the paragraph before the Data Analysis section; page 11, line 254: “there for your children” instead of “you children”; page 16, line 397: “dads play” instead of “dad’s play”)

15. Several run-on sentences should be revised (e.g., page 17, lines 402-404).

Reviewer #2: General comments:

The paper is a very nuanced and interesting characterization of paternal perceptions and views among Aboriginal men in Australia. The authors are to be commended for applying a culturally-sensitive research methodology and for bringing attention to a topic of importance and relevance to the global context. I found the focus on the researcher-informant relationship to be important and compelling, and a meaningful methodological contribution to the field of cross-cultural parenting studies. It is clear that this was a well-intended study where hard-to reach communities were given an important and voice. Despite the strength of the scope and general approach to the research, in its current form I deemed the manuscript unsuitable for publication. In this commentary I outline my main concerns and some recommendations that may help strengthen the paper.

In general, the paper reads well and is easy to follow. However, there is a need to carefully proofread for grammatical inconsistencies. At the end of each section I indicate some examples of sentences that are unclear. There is also an inconsistent use of the third person as well as the tenses used when reporting results. Some sentences use casual language, which makes the tone of the narrative heterogenous. The authors tend to include verbatim text from references, whereas it is recommendable to paraphrase and summarize the existing literature. When quoting directly, page numbers of the original source must be included.

Section-by section comments:

Introduction

The introduction does a good job of justifying the need to expand this area of research. However, I would recommend balancing the content to more broadly contextualize the study and highlight the methodological contributions of the work. Specifically, it would be helpful to identify what is known about paternal/family practices in this context, what are the main gaps, and how this research specifically addresses those research gaps. Furthermore, in my opinion the introduction would be an appropriate place to discuss the limitations of the current literature with regard to the scant cultural-sensitivity of the methods, and a good place to also discuss how this study is positioned to address the methodological limitations of the extant literature.

I recommend revising sentences such as these for grammatical consistency:

Line 58 [With regards to parenting, men in general have largely been overlooked from contributing to the

Conversation….]: revise starting paragraph with “with regards”

Line 64 [This is far from the truth]: revise – informal language

Line 80 [Although there is considerable ‘evidence demonstrating fathers’ potential to positively influence their children’s health, social success and academic achievements is now robust and compelling’]: revise grammar

Line 85 [This lack of emphasis placed on the important roles and responsibilities of Aboriginal and/or Torres Strait Islander male parents has led to regions of Australia where ‘father-son and uncle-nephew form of nurturance and authority have been eroded]’: revise grammar

Methods

In general, I would recommend transferring some of the methodological approaches (such as what is and why use yarning; the importance of researcher profiles in this line of research) to the introduction. I would recommend also streamlining the methods section to detail key processes in discrete categories: (1) sampling and recruitment; (2) demographic characteristics collected; (3) instruments; (4) data collection and analysis; (5) ethics review, consenting. I recommend providing more detail on what “community protocols” are (lime 172) and more detail on how reliability of the analysis of the qualitative data was established.

Results and Discussion

My main concerns with the scope of the work is how general the parameters of the participant sampling frame were and the absence of interpretation of the results with regard to their impact on possible programming models. With regard to the first issue, for example, the experience of parenting/fathering may vary and evolve according to the age of the children. In this study the inclusion criteria involved being a father/caregiver. I was left wondering about the crucial differences that may emerge from perceptions when exploring the caregiver experience during early childhood, mid/late childhood, and adolescence. Also, the inter-aboriginal differences that may exist tend to get diluted in the analysis. I assume there are cultural differences across the aboriginal groups explored, but by clustering the results across groups I was left with the impression that these are relatively homogeneous communities. With regard to the second issue, the authors suggest that the paper aims to inform program approaches. However, there is very limited analysis offered on the implications of the findings beyond the characterization of participant perceptions. Perhaps generating a conceptual model, or discussing the results against the backdrop of a theoretical framework, could help drive the depth of the analysis. As part of the limitations, the authors state that “Although yarning groups were conducted across urban, remote and very remote locations, due to the small sample size it may not be representative of all Aboriginal male parents across Australia”. However, in qualitative research, the aim is not to generate findings from “representative” samples, but rather apply sampling and analysis techniques that generate themes which can in turn be reliably generalized to a population/community.

Example of sentences to be revised:

Line 208: [Due to sorry business, the yarning group with Yalata participants was conducted …]: revise, please clarify

Line 220: [The men were more than willing to participate here, and it wasn’t long before we had participants who wanted to speak with us]: revise – informal language

6. PLOS authors have the option to publish the peer review history of their article (what does this mean?). If published, this will include your full peer review and any attached files.

Reviewer #1: No

Reviewer #2: No

---

## [Author Response · Author response to Decision Letter 0]

30 Oct 2019

Review Comments to the Author

Reviewer #1: This manuscript reports findings from a study with Aboriginal men residing in Australia. Specifically, the authors utilized an Indigenous cultural form of conversation (i.e., yarning) as a data-gathering tool and found several themes centering around the perceived role as a male parent as well as the motivation to be engaged in parenting programs. There are certain things to like about this study. For instance, it reports findings from a population that is not well-represented in the literature and for which our knowledge of paternal engagement in child-rearing is limited. However, there are several concerns that the authors may need to address as listed in the following paragraphs.

1. There are various direct quotes for which the page number should be added for complete and adequate reference.

Response: Some of the direct quotes have been removed while the others have had their page numbers included. 

2. Could the authors comment a bit more on the reasons that led to the lack of Torre Strait Islander males participating in any of the yarning groups? The SRG comprised both Aboriginal and Torres Strait Islander people, which should have facilitated recruitment of Torres Strait Islander.

Response: Additional text has been added to the manuscript to explain the lack of Torres Strait Islander men included in the study. “The study took place in South Australia, which has a low proportion of the population who identify as Torres Strait Islander or Aboriginal and Torres Strait Islander, 4.5% of the total Aboriginal and Torres Strait Islander population, compared to 20% of the population in Queensland (14), although both Aboriginal and/or Torres Strait Islander men were the intended participants of this study, no Torres Strait Islander men participated in any of the four yarning groups, hence the title stating ‘Aboriginal men’ only.”

3. One goal (stated in the Study Design section) was to comprehend what parents needed to strengthen their social and emotional wellbeing. Please provide an (operational) definition of social and emotional wellbeing and related questions to elicit such narratives.

Response: Additional text has been added to the study design section. “The study utilised the framework and domains of Social and Emotional Wellbeing put forth by Gee et al which depicted the domains of wellbeing that typically characterise social and emotional wellbeing from an Aboriginal and Torres Strait Islanders’ Perspective [15].”

Participants were asked “what keeps you strong as parents?” This was an example given to elicit conversation around social and emotional wellbeing from a strengthens-based approach. An additional question related to SEWB was added to the data collection section. “when it comes to caring for your child how do you lead and overcome challenges?”

4. How is “remote father” defined (Abstract)? Does “remote” refer to the study location? If so, the findings reported in the Results do not clearly describe the similarities and differences between the narratives of remote and non-remote fathers.

Response: The abstract has been modified to clarify the meaning of a ‘remote father’ “Lack of employment and therefore financial security were described as a challenge to fatherhood especially for remote fathers who live in remote communities.”

This was the only difference of note between the fathers by location, hence the differences between men by location was not discussed in the results.

5. Some information should be moved to the Data Collection section. For instance, move the information about the language in which the yarning groups were conducted (p. 9, lines 210-213) from the Results to the Data Collection section. The information about the study locations (p. 9, lines 215-227) would also be better placed in the Data Collection section.

Response: This section of text from the results section up to the data collection section.

6. Elaborate on the factors that determined data saturation. The interview guide/questions are rather limited with only two questions (“What keeps you strong as a parent?” and “What do you think you need to be successful or to continue to be successful in your role?”). Perhaps the researchers could have elicited additional information by using relevant follow-up questions. Also, how was “success” (or, being successful) defined for the second question?

Response: We agree that data saturation is difficult to prove. This statement has been removed and replaced with a statement about resource availability. “Available resources limited the study to only four yarning groups, one per community.”

There were more than two questions, the data collection section includes a list of included questions, to which we have added a question. “…when it comes to caring for your child how do you lead and overcome challenges?... 

Success was not defined by the facilitators, it was completely open for the yarning group members to interpret.

7. Which considerations guided the determination of the group size? While the groups were fairly balanced in the first three locations, there were only 5 men in the last group (Adelaide metro).

Response: The group size was not determined by the researchers, although we aimed to have somewhere between 4 and 20 men. As stated in the Data Collection section, participants were invited opportunistically.

8. It is important to present the key demographics of the participants. For instance, how old were the participants? How many were the biological fathers of the children they provided care for? How many children did the participants have and how old were they? Were all participating fathers married to the mother of their children? How many fathers were employed and other mothers employed as well? What were the household compositions and living arrangements of the participating fathers? Such information would greatly help in contextualizing the findings and themes (e.g., the finding that lack of employment and financial insecurity were described as challenges to fatherhood, or the importance of the family unit and the role of non-biological fathers). There currently are several instances in which the information would benefit from additional nuances and a more in-depth discussion of the sample characteristics. For instance, the second sentence of the Discussion states that “this study also outlines the many challenges these men face as Aboriginal men entering parenting for the first time or of those currently involved in the parenting realm.” Again, it is unclear how many men are first-time fathers and how many participants already provide care to children (and in which caregiving arrangement).

Response: We agree that demographic information from the men would have been informative for context, however, as this was the first ever exploration of fathering with Aboriginal men in South Australia, the researchers felt that the requirement to complete of a demographic form would have been a barrier to engaging in the study. In addition, many participants required assistance to understand the consent forms written in English, meaning that demographic questions would have also been required to be asked in-person by the researchers. Any information in the results section related to demographics were given freely during the group yarning.

We have however added a sentence in the limitations about this “In addition, demographic information was not collected from participants limiting our ability to determine the contextual factors that may influence their responses, such as their relationship status or their current employment status.”

9. Related to the comment above, there is inconsistency concerning the sample description. For instance, on page 5, lines 108-110, the authors clarify that no Torres Strait Islander males participated in the yearning groups. Later, on page 17, line 402, it is stated that “the Aboriginal and Torres Strait Islander male parents who participated in this research all appreciated the opportunity to participate in this research […]”. Clarify and be consistent in whether Torres Strait Islander males participated in this study. Again, a detailed description of key demographics would help in this regard.

Response: Line 402 has been modified – ‘and Torres Strait Islander’ has been deleted, as there were only Aboriginal male participants.

10. There is a reference to thematic network analyses to illustrate the relationship between the three categories. Why did the authors decide not to present these findings (“web-like map”) on basic themes, organizing themes and global themes? Doing so would help outline how the four categories (i.e., “Being a dad”, “Employment”, “Culture, country and family connectedness” and “Having an opportunity to participate in the parenting process”) were derived from the interview material.

Response: The thematic analysis has been included as a table plus some additional text at the beginning of the Results section “From the thematic analysis four global themes were identified; being a dad, employment, cultural and community connections and opportunities to participate. The analysis including the organisational and local themes are outlined in Table 1.”

11. Culture is a dominant term used throughout the Results section. To facilitate the understanding of the key findings, briefly define “culture” either based on the literature or as defined by participants who referred to culture in their narratives.

Response: A brief sentence is added to the results, under ‘culture and community connections’. 

Response: “Although cultures across Australia’s Aboriginal and Torres Strait Islander people differ greatly for example in language, customs, beliefs and traditions there are many commonalities shared, such as the importance and connection to place.”

12. The Introduction and the Discussion sections would benefit from placing the study aims and objectives into the broader research context. For instance, Michael Lamb, Joseph Pleck, and others have developed and continuously revised conceptualizations of paternal involvement. How do the findings of this study contribute to these (and related) conceptualizations?

Response: The aim of the study is not to primarily contribute to the literature, but to provide local solutions for a community identified issue and unfortunately the majority of the literature stems from research with non-Indigenous men, which the local community and researchers believe does not provide useful insights

13. Some statements and conclusions in the discussion overstretch the study findings. For instance, how can the authors be sure that the system was designed “without them in mind” (page 21, line 501)? I encourage the researchers to revise such statements according to the explanatory power of their data as well as the limitations of their study.

Response: The statement has been removed.

14. Minor proof-reading required (e.g., the paragraph before the Data Analysis section; page 11, line 254: “there for your children” instead of “you children”; page 16, line 397: “dads play” instead of “dad’s play”)

Response: These suggested edits have been made.

15. Several run-on sentences should be revised (e.g., page 17, lines 402-404).

Response: The manuscript has been revised and this example has been edited “The Aboriginal male parents who participated in this research all appreciated the invitation and opportunity to participate in this research. They were all happy to speak about the roles and responsibilities they have as parents.”

Reviewer #2: General comments:

The paper is a very nuanced and interesting characterization of paternal perceptions and views among Aboriginal men in Australia. The authors are to be commended for applying a culturally-sensitive research methodology and for bringing attention to a topic of importance and relevance to the global context. I found the focus on the researcher-informant relationship to be important and compelling, and a meaningful methodological contribution to the field of cross-cultural parenting studies. It is clear that this was a well-intended study where hard-to reach communities were given an important and voice. Despite the strength of the scope and general approach to the research, in its current form I deemed the manuscript unsuitable for publication. In this commentary I outline my main concerns and some recommendations that may help strengthen the paper.

In general, the paper reads well and is easy to follow. However, there is a need to carefully proofread for grammatical inconsistencies. At the end of each section I indicate some examples of sentences that are unclear. There is also an inconsistent use of the third person as well as the tenses used when reporting results. Some sentences use casual language, which makes the tone of the narrative heterogenous. The authors tend to include verbatim text from references, whereas it is recommendable to paraphrase and summarize the existing literature. When quoting directly, page numbers of the original source must be included.

Section-by section comments:

Introduction

The introduction does a good job of justifying the need to expand this area of research. However, I would recommend balancing the content to more broadly contextualize the study and highlight the methodological contributions of the work. Specifically, it would be helpful to identify what is known about paternal/family practices in this context, what are the main gaps, and how this research specifically addresses those research gaps. Furthermore, in my opinion the introduction would be an appropriate place to discuss the limitations of the current literature with regard to the scant cultural-sensitivity of the methods, and a good place to also discuss how this study is positioned to address the methodological limitations of the extant literature.

I recommend revising sentences such as these for grammatical consistency:

Line 58 [With regards to parenting, men in general have largely been overlooked from contributing to the

Conversation….]: revise starting paragraph with “with regards”

Response: This sentence has been revised “Men in general have largely been overlooked from contributing to the parenting conversation”.

Line 64 [This is far from the truth]: revise – informal language

Response: This sentence has since been deemed unnecessary and removed

Line 80 [Although there is considerable ‘evidence demonstrating fathers’ potential to positively influence their children’s health, social success and academic achievements is now robust and compelling’]: revise grammar

Response: This has been revised “Although there is considerable ‘evidence demonstrating fathers’ potential regarding the positively influence father can have on their children’s health, social success and academic achievements is now robust and compelling’ [10], the roles and responsibilities of male parents continue to be neglected within Australian maternal and early years services and research.”

Line 85 [This lack of emphasis placed on the important roles and responsibilities of Aboriginal and/or Torres Strait Islander male parents has led to regions of Australia where ‘father-son and uncle-nephew form of nurturance and authority have been eroded]’: revise grammar

Response: The grammar of this sentence has been revised and moved up into the previous paragraph. “The lack of emphasis placed on the important roles and responsibilities of Aboriginal and/or Torres Strait Islander male parents has led to regions of Australia where father-son and uncle-nephew relationships have been eroded [11]”

Methods

In general, I would recommend transferring some of the methodological approaches (such as what is and why use yarning; the importance of researcher profiles in this line of research) to the introduction. I would recommend also streamlining the methods section to detail key processes in discrete categories: (1) sampling and recruitment; (2) demographic characteristics collected; (3) instruments; (4) data collection and analysis; (5) ethics review, consenting. I recommend providing more detail on what “community protocols” are (lime 172) and more detail on how reliability of the analysis of the qualitative data was established.

Response: The methodological approaches have been deemed to not fit within the Introduction and have remained in the Methods section. Additional categories have been added to the Data Collection section of the manuscript as suggested. Additional text has been included to provide more information on ‘community protocols’ “All participants were invited to participate opportunistically at locations deemed suitable by the participants and at suitable times according to community protocols; for example, avoiding cultural ceremony or sorry business (death in the community).” Reliability is outlined in the Data Analysis section. No additional reliability methods were used.

Results and Discussion

My main concerns with the scope of the work is how general the parameters of the participant sampling frame were and the absence of interpretation of the results with regard to their impact on possible programming models. With regard to the first issue, for example, the experience of parenting/fathering may vary and evolve according to the age of the children. In this study the inclusion criteria involved being a father/caregiver. I was left wondering about the crucial differences that may emerge from perceptions when exploring the caregiver experience during early childhood, mid/late childhood, and adolescence. Also, the inter-aboriginal differences that may exist tend to get diluted in the analysis. I assume there are cultural differences across the aboriginal groups explored, but by clustering the results across groups I was left with the impression that these are relatively homogeneous communities. With regard to the second issue, the authors suggest that the paper aims to inform program approaches. However, there is very limited analysis offered on the implications of the findings beyond the characterization of participant perceptions. Perhaps generating a conceptual model, or discussing the results against the backdrop of a theoretical framework, could help drive the depth of the analysis. As part of the limitations, the authors state that “Although yarning groups were conducted across urban, remote and very remote locations, due to the small sample size it may not be representative of all Aboriginal male parents across Australia”. However, in qualitative research, the aim is not to generate findings from “representative” samples, but rather apply sampling and analysis techniques that generate themes which can in turn be reliably generalized to a population/community.

Response: There are significant cultural differences across Aboriginal and Torres Strait Islander people, but also commonalities. A sentence was added to under ‘Culture and community connections’ – “Although cultures across Australia’s Aboriginal and Torres Strait Islander people differ greatly for example in language, customs, beliefs and traditions there are many commonalities shared, such as the importance and connection to place.”

Example of sentences to be revised:

Line 208: [Due to sorry business, the yarning group with Yalata participants was conducted …]: revise, please clarify

Response: Sorry business has been clarified “…sorry business (death in the community).”

Line 220: [The men were more than willing to participate here, and it wasn’t long before we had participants who wanted to speak with us]: revise – informal language

Response: This sentence has been revised “The men were more than willing to participate at this setting, with 12 participants joining the yarning group.”

---

## [Editor Report · Decision Letter 1]

5 Nov 2019

“Anybody can make kids; it takes a real man to look after your kids”: Aboriginal men’s discourse on parenting.

PONE-D-19-21777R1

Dear Dr. Canuto,

We are pleased to inform you that your manuscript has been judged scientifically suitable for publication and will be formally accepted for publication once it complies with all outstanding technical requirements.

With kind regards,

Aisha K Yousafzai

Academic Editor

PLOS ONE
---

## [Editor Report · Acceptance letter]

15 Nov 2019

PONE-D-19-21777R1 

“Anybody can make kids; it takes a real man to look after your kids”: Aboriginal men’s discourse on parenting. 

Dear Dr. Canuto:

I am pleased to inform you that your manuscript has been deemed suitable for publication in PLOS ONE. Congratulations! Your manuscript is now with our production department. 

With kind regards,

on behalf of

Dr. Aisha K Yousafzai 

Academic Editor

PLOS ONE